# Biological Cloth Face Coverings—The Reduction of SARS-CoV-2 and Influenza (H1N1) Infectivity by Viruferrin™ Treatment

**DOI:** 10.3390/ma14092327

**Published:** 2021-04-30

**Authors:** Emily S. Medina-Magües, Anna Stedman, Paul Hope, Jorge E. Osorio

**Affiliations:** 1School of Veterinary Medicine, University of Wisconsin-Madison, 1656 Linden Dr., Madison, WI 53706, USA; esrussell@wisc.edu; 2Virustatic Ltd., M-Sparc Parc Gwyddoniaeth Menai, Menai Science Park, Gaerwen, Isle of Anglesey, Wales LL60 6AG, UK; annastedman@virustatic.com (A.S.); paulhope@virustatic.com (P.H.); 3Colombia Wisconsin One Health Consortium, Universidad Nacional Medellín, Calle 75#79a 5, Medellín 050034, Colombia

**Keywords:** COVID-19, face masks, personal protective equipment, personal protective material, droplet, aerosols

## Abstract

In an attempt to create novel methods to reduce the transmission of SARS-CoV-2 and influenza viruses, fabric material was treated with Viruferrin™ and tested for its inactivating properties against the pandemic severe acute respiratory syndrome coronavirus 2 (SARS-CoV-2) and influenza A viruses. Inactivating properties were evaluated by comparing Viruferrin-treated and cotton control fabric material with and without the application of saliva at various time points after virus exposure. A statistically significant (*p* < 0.0001) decrease in the number of infectious virus particles exposed to Viruferrin-treated fabric when compared with the cotton control for both SARS-CoV-2 and influenza A viruses was observed. For both SARS-CoV-2 and influenza A, Viruferrin-treated fabrics experienced a >99% virus reduction without saliva after five minutes of contact when compared to the positive control at time point 0. Furthermore, the reusability of the Viruferrin treated fabric was demonstrated by stability for up to 10 washes. The level of anti-viral (SARS-CoV-2) activity remained constant from 5 to 10 washes and demonstrated a significant difference (*p* < 0.0001) from the unwashed untreated material. Applications for this treated fabric are far reaching, as a biological face covering offers not only a unique 2-way protection but also is unlikely to cause onward touch transmission.

## 1. Introduction

Respiratory pathogens can cause a significant global health burden in humans and turn into major pandemics. For example, SARS-CoV-2, the etiologic agent of Coronavirus disease of 2019 (COVID-19), has evolved into a global pandemic and, as of 14 January 2021, resulted in the death of over two million individuals and at least 100 million cases [1]. The recovery from this infection can also result in long-term effects, including lung function abnormalities, acute kidney injury, effects to mental health, and cardiovascular and neurological symptoms [2]. Prior to the emergence of SARS-CoV-2, various strains of influenza viruses have circulated at the pandemic level. The most recent influenza pandemic emerged in 2009, named influenza A H1N1pdm09 virus. It is estimated that between 2009–2018, this virus has caused at least 100 million cases, 900,000 hospitalizations, and 75,000 deaths [3]. Influenza viruses can also circulate in seasonal epidemics which the WHO estimates are the cause of 290,000–650,000 deaths each year [4]. There is a concern that, as the world recovers from the pandemic, new influenza viruses may evolve with higher mortality and morbidity, and with a reduced immunity in the population due to stringent social distancing measures [5].

The occurrence of these pandemics has also led to significant global economic, social, and political disruptions that have highlighted the need for strategies to reduce their spread. For COVID-19, because of the global rush to procure vaccine doses, as well as the frequent emergence of SARS-CoV-2 variants, many countries have utilized public health interventions to reduce the risk of community transmission. They have included border control or closure, quarantine and testing of all incoming travelers, reverse transcription polymerase chain reaction (RT-PCR), loop-mediated isothermal amplification (LAMP) and lateral flow assays testing for case detection, rapid contact tracing and quarantine, frequent hand hygiene, face masks, social distancing measures including school closures, home office, cancellation of all mass gatherings, stay-at-home orders, curfews, and the cessation of many socioeconomic activities.

While the use of face masks or facial coverings has remained controversial for certain community sectors, there are some transmission characteristics of COVID-19 that highlight their importance. They include (1) viral spread through small droplets even while talking; (2) a significant proportion of asymptomatic cases; and (3) a long pre-symptomatic incubation period (up to 15 days) during which viral transmission occurs [6,7]. There is evolving evidence that the virus is not only droplet spread (when an individual coughs, sneezes) but that it can be transmitted by aerosols when speaking (for field transmission) [8]. Furthermore, recent studies indicate that widespread community mask use can significantly reduce viral transmission [9]. Due to the explosive transmission nature of the current COVID-19 pandemic resulting in a global scarcity and price soaring of fluid resistant surgical masks (FRSM) and N95 respirators, governments have been prompted to recommend face covers made of any available cloth material. However, the filtering capacity of fabric materials for small particles (>0.1 um in size) can be as low as 5% [10].

In an attempt to discover materials that reduce the transmission of SARS-CoV-2 and influenza viruses, different materials were screened for antiviral properties and selected for fabric application. After screening, fabric material was treated with Viruferrin™ containing bovine lactoferrin which is previously demonstrated as a preventative against SARS-CoV-2 infection and reduces the disease duration and severity in patients with SARS-CoV-2 [11]. Viruferrin-treated fabric was tested for its inactivating properties and durability when applied to fabric. Health mark artificial saliva test soil, which mimics the protein content and constitution of saliva, was used to evaluate the performance of the Viruferrin-treated and cotton controls while imitating natural respiration when wearing a face mask. Furthermore, the reusability of the Viruferrin treated fabric was evaluated for stability under standard laundering conditions for up to 10 washes. It should be noted that further testing may extend the maximum number of washes.

## 2. Materials and Methods

### 2.1. Viruses and Cells

For the Material Testing Part 1, African Green Monkey kidney cells (Vero E6, ATCC #CRL-1586, Manassas, VA, USA) and Madin-Darby Canine Kidney cells (MDCK [NBL-2], ATCC #CCL-34, Manassas, VA, USA) were propagated in Dulbecco’s modified Eagle medium, (DMEM; Gibco, Carlsbad, CA, USA) augmented with 2–5% fetal bovine serum (FBS) and antibiotics and incubated at 37 °C in 5% CO_2_. Two viruses obtained from BEI resources (NIAID, NIH, Manassas, VA, USA) were used: SARS-CoV-2 (USA-WA1/2020 isolate, ref# NR-52281, Manassas, VA, USA) and H1N1 (Influenza A Virus, A/California/04/2009, ref# NR-13658, Manassas, VA, USA). Virus stocks were tittered using TCID₅₀ (Median Tissue Culture Infectious Dose) following established procedures [12].

For the Materials Testing Part 2, Vero E6 cells (ATCC #CRL-1586, Manassas, VA, USA) were maintained in Eagle’s minimum essential medium (EMEM; Lonza, Milano, Italy) supplemented with 2 mM L- Glutamine (Lonza, Milano, Italy), 100 units/mL penicillin-streptomycin mixture (Lonza, Milano, Italy) and FBS (Euroclone, Pero, Italy) to a final concentration of 5%, at 37 °C, in a 5% CO_2_; humidified incubator. SARS-CoV-2 was obtained from the European Virus Archive global (EVAg, Marseille, France).

### 2.2. Cell Sensitivity Studies

In order to confirm that the suppression of viral activity is due to the Viruferrin and not any other factor, such as leaching from the fabric material, the lack of cell toxicity and the cells’ sensitivity to the virus was established by assessing the impact of the wash out solution on the cells, and hence the viral infectivity. The virus was incubated in the wash out solutions and the viral loads compared between the Viruferrin-treated fabric and the cotton control wash out solution. This would ensure that any observed reduction in viral titer in the actual test would be related to the applied treatment. The wash-out volume was also compared, between replicates and between fabrics, to ensure consistency and to prevent any bias occurring in the experimental design.

Three untreated control samples (plain cotton fabric) and three Viruferrin-coated samples (Viruferrin-coated fabric) were prepared verifying equal size (20 mm × 20 mm) and weight (0.40 g). Control samples were prepared in accordance with established procedures [12]. In brief, sample vials with prepared fabric were mixed with 500 µL of wash out solution (DMEM) by vortexing each 5 times for 5 s. The wash out solution was transferred to a new sterile container prior to adding either 50 µL of SARS-CoV-2 or 50 µL of H1N1 at a concentration of 1.0 × 10⁷ plaque forming units (PFU)/mL. The tubes were incubated for 30 minutes (min) at 25 °C and then serial dilutions were performed to determine the infectious titer by TCID_50_-50% infectious dose of a wash out virus suspension or the dilution of the virus suspension that induces a cytopathic effect (CPE) in 50% of cell culture units.

### 2.3. Materials Testing Part 1

Viruferrin™ is a proprietary treatment for fabrics. Viruferrin™ is composed of PROFERRIN^®^ (Ingredia, Arras Cedex, France), a bovine lactoferrin in spray-dried powder with >95% purity.

In the first study a total of 160, control and treated fabric, samples were used. Eighty Viruferrin treated samples were evaluated for each virus. Two sets of testing (with and without saliva) were conducted, each with two samples per time points (0, 1, 5, 15, 30, 60, 120, 360, 720, and 1440 min). The same number of samples and time points were used for the control material. A positive control for infection—media and virus with no fabric—was included.

Sample materials (plain cotton control fabric and Viruferrin-coated fabric) were prepared verifying equal size (20 mm × 20 mm) and weight (0.40 g) in accordance with previously established protocols [12]. Fabric samples were either coated dropwise with 500 µL of reconstituted health mark saliva test soil (Fraser, MI, USA, ref# STS-100ML) onto one side of the fabrics to simulate saliva and breath on the inside of the mask or left without saliva. Fabric samples were allowed to dry and then placed into vials where 200 µL of 1 × 10⁷ PFU/mL of either SARS-CoV-2 or H1N1 virus was applied dropwise onto one side of the fabric. For samples with saliva, virus was applied to the side of the fabric without the saliva test soil. Sample vials were sealed and incubated at room temperature for the appropriate contact times. Following incubation, samples were resuspended in 500 µL of media (DMEM) and vortexed 5 times for 5 s. The eluted fluid was then used immediately to infect Vero E6 cells (SARS-CoV-2) or MDCK cells (H1N1) according to the procedure below.

### 2.4. Materials Testing Part 2: Reusability and Stability of the Viruferrin Treatment

SARS-CoV-2 was prepared using 200 μL of 1.58 × 10^7^ TCID₅₀/mL SARS-CoV-2 diluted in 2mL of media and evenly spread into individual wells on a 9-well plate. For the washed materials, the Viruferrin treated fabric material was washed at 30 °C with non-biological washing powder (Persil) in a standard household washing machine (Indesit) on a 9-min cycle. This was repeated 5 and 10 times. As a control, the same fabric material untreated and unwashed was used. Equal sized pieces of fabric material were prepared, and the test was conducted in triplicate. Each individual inoculated well containing SARS-CoV-2 was wiped, with a single action, per piece of fabric. The viral titer of each well was determined using the microneutralization CPE-based assay thereby enabling for the amount that was transferred on to the fabric material to be calculated.

### 2.5. Quantification of Viral Load

For the Materials Testing Part 1 Vero E6 cells were seeded at 10,000 cells/well into a 96-well plate (Sigma-Aldrich, Darmstadt, Germany) 48 h pre-infection and incubated at 37˚C and 5% CO_2_ while MDCK cells were seeded at 60,000 cells/well into a 96-well plate 24 h pre-infection. The virus recovered from the test fabric was serially diluted in a mixing plate in duplicate and added to the 96-well plates. Plates were incubated for 1 h at 37 °C and 5% CO_2_. Following incubation, a carboxymethylcellulose overlay was added. Plates were then incubated for 36 h at 37 °C and 5% CO_2_. The overlay was discarded. Plates were fixed for 10 min at −20 °C with an acetone, methanol, and acetic glacial acid solution. Following fixation, plates were washed two times with TBS-T and the primary antibody (SARS-CoV-2: Monoclonal Anti-SARS Coronavirus Recombinant Human IgG1, Clone CR3022 (Abcam, Cambridge, UK, abID# ab273073); H1N1: Monoclonal antibody Anti-NP protein, Influenza a Virus, Clone 2F4 (BEI Resources, NR-19868, Manassas, VA, USA)) was added and the plates were incubated overnight at 4 °C. Excess primary antibody was then discarded by washing the plates twice with TBS-T. The secondary antibody (SARS-CoV-2: Goat Anti-human IgG H&L HRP Conjugated; H1N1: Goat Anti-Rabbit IgG HRP conjugated) was added and left to incubate for 2 h at 37 °C. The plates were then washed twice with TBS-T and plaques were developed with a Chromogen/Peroxidase substrate. Plaques were counted using an open-source software Viridot. Plaques were averaged between replicates and the titer was calculated following established protocols that maintained a limit of detection below 10^3^ PFU [12].

For the Materials Testing Part 2, a microneutralization CPE-based assay was performed to quantify the amount of viable virus. Each dilution, in duplicate, was used to infect Vero E6 cells and incubated for 72 h at 37 °C with 5% CO_2_. The primary endpoint was determined by checking for the presence of the cytopathic effect of the virus on the cells using microscopy. The last dilution of the compound showing 50% of CPE over the cell layer was considered to be the EC50 value. These tests were performed according to the Standard Operating Procedures in place at VisMederi, the University of Siena under Good Laboratory Practice (GLP) requirements. The virus was back titrated using the microneutralization CPE-based assay to determine the initial starting titer.

### 2.6. Data Analysis

For the Materials Testing Part 1, cell sensitivity to treatment was measured by comparing the geometric mean titer (GMT) of three plain cotton control fabric samples to three Viruferrin-treated fabric samples to ensure Viruferrin-treated fabric wash out solution was negative for cytotoxicity and that the wash-out solution was of a sufficient volume to ensure accurate measurements between treatment and control groups for both H1N1 and SARS-CoV-2. GMTs were evaluated by ensuring the difference between the reference specimen or the control and the antiviral specimen or the Viruferrin-treated fabric was less than or equal to 0.5. Any values greater than the ratio of 0.5 required an increase in the volume of the wash-out solution as stated in previous protocols [12]. Once the volume of the wash out solution was verified as an accurate measurement between treatment and control groups in the cell sensitivity study, the volume was then applied to quantify the viral load in virus inactivation testing. Percent reduction of geometric mean viral titers were calculated by comparing the plain cotton control at time 0 and the Viruferrin-treated fabric following previously established protocols [13]. Antiviral activity values were calculated comparing the average control titer immediately after inoculation at time point 0 compared to the average treated titer at various time points from 0 to 1440 min established by previous protocols [12]. Statistical analysis was performed (GraphPad version 8.0 software: La Jolla, CA, USA). Normally distributed continuous data (i.e., viral titers) was assessed using a 2-way ANOVA.

For the Materials Testing Part 2, the GMTs were calculated from the microneutralization CPE-based assay and compared by 2-way ANOVA, with Sidak’s multiple comparisons test, for the viral transfer assays (GraphPad version 8.0 software: La Jolla, CA, USA).

## 3. Results

### 3.1. Cell Sensitivity Studies

Infectious titers were determined by TCID₅₀ for both H1N1 and SARS-CoV-2 using three control cotton fabrics and three Viruferrin coated fabric samples to test for cell sensitivity to treatment in accordance with previously established protocols [12]. GMTs for H1N1 and SARS-CoV-2 (depicted in Equations (1) and (2), respectively) that were eluted from both the control and Viruferrin-treated fabric had identical GMTs indicating that Viruferrin treatment did not induce cell toxicity and that the wash-out solution was sufficient in volume to ensure accurate measurements.
(PFU/mL of reference specimen) − (PFU/mL of antiviral specimen) ≤ 0.5(6.32 × 10³ PFU/mL cotton) − (6.32 × 10³ PFU/mL treated) ≤ 0.50 ≤ 0.5(1)
(PFU/mL of reference specimen) − (PFU/mL of antiviral specimen) ≤ 0.5(7.76 × 10^4^ PFU/mL cotton) − (7.76 × 10^4^ PFU/mL treated) ≤ 0.50 ≤ 0.5(2)

### 3.2. H1N1 Inactivation by Viruferrin Treated Fabric (Material Testing Part 1)

Viruferrin-treated and untreated cotton fabric samples were incubated with H1N1 virus over a time course of 0 to 1440 min and the eluted virus was tittered by plaque assay. Differences between GMTs and standard deviations indicated in Figure 1A,B were analyzed using a 2-way ANOVA. H1N1 GMTs of Viruferrin treated fabrics without saliva showed a statistically significant difference in viral titer for all timepoints up to 120 min (*p* ≤ 0.0001) and at 360 min (*p* = 0.0007), with detectable viral titers, when compared to the cotton controls, whereas, with saliva there was a significant difference at 0-, 1- and 5-min timepoints with *p* values of <0.0001, <0.0001 and 0.01, respectively.

Virus percent reduction was calculated comparing treatment with the positive control at timepoint zero with and without saliva in Figure 2A. An initial reduction of 92.59% and 98.92% at timepoint 0- and 1-min, respectively, was observed with fabric samples without saliva. After 1 min of Viruferrin-treatment with saliva, a 75.24% reduction in viral titer was calculated and at 5 min this increased further to 93.16%. After 60 min of contact time with Viruferrin-treated fabric there was a 99.8% viral reduction with saliva and undetectable levels of virus at 120-min. However, without saliva there was a 99.45% reduction in viral titer after 5 min of contact with the Viruferrin treated fabric.

Antiviral activity values were calculated for samples with and without saliva in Figure 2B by comparing GMTs eluted from Viruferrin-treated fabric at various timepoints to GMTs eluted from cotton controls at timepoint zero. Samples without saliva showed an initial antiviral activity value at timepoint zero of 0.4268 and reached 1.559 at 5 min. Antiviral activity values without saliva further increased with exposure time and reached 2.79 after 120 min of Viruferrin-treated exposure. The initial antiviral activity value for samples with saliva reached 0.2779 at timepoint zero and increased to 1.501 at 15 min of exposure. Antiviral activity values with saliva increased with longer fabric exposure times and reached 2.903 after 120 min of Viruferrin-treated exposure.

### 3.3. SARS-CoV-2 Inactivation by Viruferrin Treated Fabric (Material Testing Part 1)

Viruferrin-treated and untreated cotton fabric samples were incubated with SARS-CoV-2 virus from 0 to 1440 min and eluted virus was tittered using plaque assay. SARS-CoV-2 GMTs of treated fabrics showed a statistically significant difference in viral titer for timepoints 0 and 1 min (*p* ≤ 0.0001) when compared to cotton controls (Figure 1B). There was a reduction in viral titer for all timepoints in the presence of saliva when compared to the cotton control, although, this was not deemed to be significant (Figure 1A).

An initial virus reduction, when compared to the positive control at timepoint zero, of 23.15% and 83.1% for samples with and without saliva, respectively, was observed (Figure 2C). This reduction rapidly rose to 99.4% at 5 min without saliva and to 96% at 15 min with saliva. Increased exposure to Viruferrin-treatment in samples led to undetectable levels of virus above 360 min.

Antiviral activity was calculated by comparing the GMT of SARS-CoV-2 eluted from Viruferrin-treated fabric at various timepoints to virus eluted from cotton controls at timepoint zero. Initial antiviral activity values at timepoint zero for samples without saliva were 0.4466 and reached a value of 2.055 after 15 min of exposure time (Figure 2D). Antiviral activity values increased with longer exposure times and samples without achieved a value of 3.140 after 360 min of Viruferrin-treated exposure. After 15 min, antiviral activity values for samples with saliva reached 1.308 and continued to increase upon further fabric exposure with a final value of 2.838 after 720 min of Viruferrin-treated exposure.

### 3.4. The Ability of Viruferrin Treated Fabric to Inactivate SARS-CoV-2 after Washing up to Ten Times (Material Testing Part 2)

There was a significant reduction in viral titer with the washed fabrics when compared to the unwashed and untreated control material (*p* = 0.0004) (Figure 3A). However, there was no significant difference between washing the fabric material either 5 or 10 times (Ordinary one-way ANOVA with Tukey’s multiple comparisons test). The percentage of viral capture was 99.9% for both the 5×- and 10×-washed Viruferrin treated fabric materials whereas it was 82% for the unwashed and untreated control fabric material (Figure 3B).

## 4. Discussion

Respiratory pandemic viruses such as influenza and SARS-CoV-2 are frequently transmitted by airborne droplets and droplet nuclei from infected people through breathing, speaking and coughing. The use of fabric face masks by the general public to reduce viral spread have become commonplace after the most recent SARS-CoV-2 pandemic. However, the filtration efficiencies for fabrics can range from 5 to 80% for particle sizes of >300 nm [10]. For masks currently available to the general public from bandanas to commercially bought masks, fitted filtration efficiencies vary between 26.5% and 79.0% for particle sizes between 0.02–0.60 μm [14]. Also, gaps caused by an improper fit of standard masks causing sideways leakage, can result in over a 60% decrease in the filtration efficiency [10]. In addition, the impact on the environment by the generated PPE waste has yet to be fully determined.

Here, we demonstrate the application of a Viruferrin treatment to a breathable fabric material, which has subsequently been used as a biological cloth face covering (BCFC), namely the Virustatic Shield. This Virustatic Shield uses a snug face fit eliminating the decrease in viral filtration caused by sideways leakage of other common face masks. The authors have in addition positively evaluated the BCFC dermatology for skin irritation and skin sensitization using a human repeat insult patch test (BioScreen Testing Services, Torrance, California) which demonstrated that there was no indication of a potential to elicit dermal irritation or sensitization (contact allergy) (data not shown) [15].

Viruferrin, comprising of bovine lactoferrin, is a natural immunomodulatory iron-binding glycoprotein with strongly cationic properties. Lactoferrin has a generally regarded as safe (GRAS) status by the Food and Drug Administration (FDA) with no contraindications in either pediatric or adult patients. Lactoferrin is a component of many commercial products, including that of babies’ milk formula. As seen in other in vitro studies, lactoferrin, which is expressed in most biological fluids, has been shown to inhibit a variety of pathogens including influenza A and a virus closely related to SARS-CoV-2, the Severe Acute Respiratory virus (SARS-CoV) from 2003 [16]. In this paper, we have demonstrated that lactoferrin is also capable of inhibiting SARS-CoV-2 when used as an antiviral application on fabric material. While the exact mechanism for inhibition of SARS-CoV-2 by lactoferrin is unknown, it has been shown to inhibit viral entry for related-virus SARS-CoV either through virus-binding or cell surface molecule binding [16,17,18,19]. In clinical trials where lactoferrin is used as an antiviral, it has been shown to reduce the severity and duration of disease in patients with SARS-CoV-2 within 4–5 days after oral administration and prevent contraction of SARS-CoV-2 in 256 people (100% of participants) with contact with these infected symptomatic individuals [20]. These trials suggest a previously noted mechanism of action of lactoferrin against the infectivity of SARS-CoV-2 which agrees with the in vitro viral inactivation of SARS-CoV-2 generated in this study.

In this study we found significant virus reduction against both H1N1, with and without the application of saliva, and SARS-CoV-2, without saliva, when comparing Viruferrin-treated fabric and plain cotton fabric. For both viruses, exposure to Viruferrin-treated fabric without saliva resulted in >99% reduction in viral titers as early as 5-min after treatment. Although the impact of saliva was assessed here and despite having a significant effect on the anti-viral capabilities for Viruferrin (a reduction in effectiveness from 92.6% to 33% with H1N1 and from 83% to 23% with SARS-CoV-2, both upon immediate contact), the authors believe that, for the general public, users of the face covering will not be exposed to significant volumes of saliva such that it would reduce the capability of the face covering. However, that said, as the contact time increased this difference in viral reduction was minimized and eventually the effect was fully overcome. The non-treated materials have also been demonstrated to have an effect on the viability of the virus. This is believed to be due to the loss of virus during the process of absorption and elution from fabric material in addition to the natural decrease in virus viability over time.

For both H1N1 and SARS-CoV-2, viral particles were undetected, using a limit of detection below 1000 PFU, after 60 and 120 min and 720 and 360 min, with and without saliva. For Influenza, the minimum infectious dose is estimated to be around 700 PFU and while studies determining the SARS-CoV-2 minimum infectious dose are ongoing, it is estimated to be around 300 PFU using computational modeling [21]. Although the limit of detection of this assay is above the minimum infectious dose required for H1N1 and SARS-CoV-2 infection, the drop in viable virus particles will decrease the potential exposure and infection during mask handling and disposal.

The stability of the Viruferrin treatment on the fabric material was tested and found to be stable, here for up to 10 washes. Further testing is required to determine the longevity of the treatment but the reusability of the BCFC should reduce the cost to consumers when compared with single-use face masks. The environmental impact from the usage of single-use PPE during the 2020–2021 SARS-CoV-2 pandemic has not yet been fully assessed, however, it is believed to be of catastrophic proportions. Not only does it present a biohazard, but it is non-biodegradable. The emergence of a reusable innovative BCFC will help to re-address the balance and hence limit the accumulation of contaminated wastage from disposable face coverings as it can be disposed of in a normal textile recycling facility and is not considered biological waste.

The dual-purpose effect of the BCFC protects the wearer from others and also protects others from an infected wearer, which is extremely significant given the high number of asymptomatic COVID-19 people who are unaware that they are shedding the virus and contributing to the community spread.

Thus, treated face mask fabric materials can provide an additional mechanism of protection by disrupting the transmission of viruses and therefore the number of cases and fatalities resulting from these infections. Decreasing infectious virus particles were seen upon increased exposure time to the treated fabric materials suggesting that Viruferrin works to decrease the amount of virus particles capable of infecting a susceptible host. For the general public, treated face mask fabric material can further decrease the rate of infection of respiratory diseases by inactivating virus particles upon contact with the treated fabric. While we tested Viruferrin for Influenza A and SARS-CoV-2 viruses, additional respiratory viruses should be evaluated for virus reduction. In addition to being available to the general public for protection against respiratory diseases, antiviral treated fabric material can be further incorporated for the use in surgical masks and different fabrics common in healthcare settings and waiting rooms.

## Figures and Tables

**Figure 1 materials-14-02327-f001:**
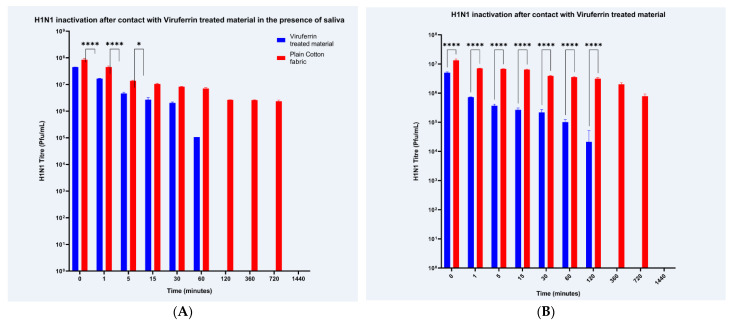
Inactivation of H1N1 and SARS-CoV-2 when in contact with a Viruferrin treated fabric material. 20 mm × 20 mm pieces of treated (Viruferrin-treated) and untreated (Cotton) control fabric samples, with (**A**,**C**) and without (**B**,**D**) the application of saliva, were assessed for anti-viral capabilities at 10 timepoints over a total time period of 1440 min with a limit of detection below 10^3^ PFU. The bars represent geometric mean titers and the standard deviations are shown with error bars. Data was analyzed for significance using a 2-way ANOVA with Sidak’s multiple comparisons test (alpha = 0.05, *p* **** ≤ 0.0001 with * *p* value = 0.0162) comparing Viruferrin-treated and cotton control fabric samples at each timepoint.

**Figure 2 materials-14-02327-f002:**
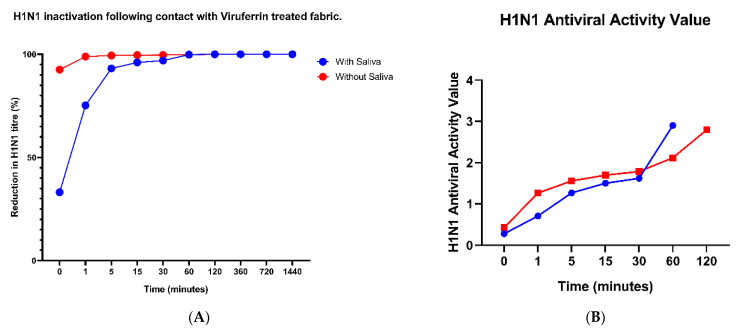
(**A**) Accumulated percentage reduction and (**B**) antiviral activity values of H1N1 and the (**C**) accumulated percentage reduction and (**D**) antiviral activity value of SARS-CoV-2 from Viruferrin-treated fabric samples collected at 10 time points (0 to 1440 min or 24 h). Virus percent reduction was calculated in relation to virus recovered from (**A**,**C**) the positive control while antiviral activity values (**B**,**D**) were calculated in relation to the cotton material control at timepoint 0, with and without the application of saliva. Antiviral activity values after 60 and 120 min, with and without saliva, respectively for H1N1 and 720 min for SARS-CoV-2 are not plotted as they are zero and it is necessary to calculate using the logarithmic values.

**Figure 3 materials-14-02327-f003:**
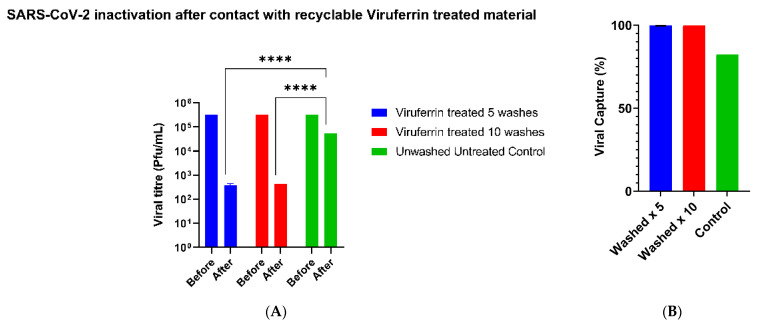
The stability of the Viruferrin treated fabric material was assessed by washing the treated fabric material either 5 or 10 times. Each assay was conducted in triplicate with an unwashed non-treated fabric as control. Each individual inoculated well containing SARS-CoV-2 was wiped, with a single action, per piece of fabric. The viral titer of each well was determined using the microneutralization CPE-based assay. (**A**) The bars represent GMTs, and the standard deviations are shown with error bars. Data was analyzed for significance using a 2-way ANOVA with Sidak’s multiple comparisons test (alpha = 0.05, *p* **** ≤ 0.0001) comparing Viruferrin-treated and the control fabric samples. (**B**) The Capture Rate of SARS-CoV-2 is expressed as a percentage based upon the viral titer of the starting inoculum. Each bar represents the test in triplicate.

## Data Availability

The data presented within this study are available within the article.

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
