# Peer review of "Biological Cloth Face Coverings—The Reduction of SARS-CoV-2 and Influenza (H1N1) Infectivity by Viruferrin™ Treatment"

_materials, 2021, doi:10.3390/ma14092327_

Round 1

Reviewer 1 Report

Medina-Magues, E. et al. have studied the effect of Viruferrin-treated cloth masks to inactivate virus particles. This study is timely, and has potential to be a viable alternative to the surgical and N95 masks whose disposal after the use is of great concern. The authors also clearly demonstrate the repeated washings (up to 10 times) did not have any noticeable decrease in the efficacy of viral capture.  There are a few minor points may be considered prior to publication.

  • Figure 3B resolution needs to be improved and should have error bars.
  • Legends of Figure 1 and 2 are somewhat unclear. C and D should explicitly be explained.
  • A sentence regarding how authors have come up with the idea of using Viruferrin as a virus inactivating agent will be highly helpful. Are the authors inspired by any previous works?

Reviewer 2 Report

In this manuscript, the authors were attempted to create novel methods to reduce the transmission of SARS-CoV-2 and influenza viruses. For these purposes, they tested the fabric material coated with ViruferrinTM, and it was confirmed that Viruferrin-treated fabrics experienced a>99% virus reduction, and also it was checked the reusability of Viruferrin-treated fabrics. The theme of this manuscript is considered quite timely and important.

However, I feel the point of the argument was not clear. It was not clear why Viruferrin-treated fabrics was selected for this purpose. It was apparent in “Discussion”. The paragraph L.304-L.320 should be described in Introduction. In this paragraph, it was well explained that why Viruferrin was important.

Furthermore, I have a question in Fig. 1. Inactivation of H1N1 and SARS-CoV-2 when in contact with a Viruferrin-treated fabric material. The virus reduction was suddenly occur after 120, 240, 1440, and 740 min, in A, B, C, and D, respectively. These value were analyzed by analysis of variance. I want to know why the sudden drop occurred.

It was referred in this manuscript that the further washing. Is it necessary to commercialize the Viruferrin-treated fabrics?

Reviewer 3 Report

The study is very interesting, and the results could potentially be of high significance in the time of the COVID-19 pandemic and more.

Nevertheless, I have some comments to the presentation of the study.

Title

“Biological Cloth Face Coverings - The reduction of SARS-CoV-2 and Influenza (H1N1) infectiv-3 ity by ViruferrinTM treatment”

I suggest considering another term than "treatment"

Abstract

I suggest supplementing the abstract with a specification of the study purpose and the research method.

  1. line

“severe acute respiratory syndrome 2 (SARS-CoV-2)”

  • incorrect term

Introduction

Reference [1]

Requires checking the citation metric.

Reference [3]
Unclear metric, no source identification

  1. Line

“(Center for Disease Control and Prevention, 2019). ”

The source of the information has been omitted

Line 38. ”There is a concern that as the world recovers from  the pandemic that new influenza viruses may evolve with higher mortality and morbidty and with a reduced immunity in the population due to stringent social distancing ”

Information without the source

  1. Line

”For COVID-19, because of the global lack of accessible vaccines ”

Information requires updating.

  1. Line

”RT-PCR, LAMP” 

Abbreviations used for the first time require explanation

  1. Line

”a long pre-symptomatic incubation period (up to 15 days) during which viral transmission occurs”

I recommend to clarify the time of viral transmission

Results

3.2.

Line 219

“An initial reduction of 92.59% and 98.92 % at timepoint 0- and 1-minute, respectively was observed with fabric samples without saliva.”

Figure 1. does not illustrate the reduction in viral load of 92 and 98% over this time period (both 107- 108).

I suggesting providing an explanation

Line 247.

“This reduction rapidly rose to 99.4% at 5 minutes without salivaand to 96% at 15 minutes with saliva.”

A similar remark as above.

Line 255.

“After 15 min, antiviral activity values for samples with saliva reached 1.308 and continuedto increase upon further fabric exposure with a final value of 2.838 after 720 minutes of Viruferrin-treated exposure (Figure 4).”

I did not find figure 4.

Line 248

“Increased exposure to Viruferrin-treatment in samples led to undetectable levels of virus above 360 minutes.”

I recommend explaining the practical significance of this result (in the discussion). How long, according to the authors, does the tested mask ensure safety after exposure to SARS-CoV-2 virus at point "0"? Can the virus persist in the mask for 360 minutes (6 hours)? In my opinion, this is the most important missing point in the discussion.

Discussion

288 -295. line

The information is a duplicate of the introduction and may be omitted.

  1. line ”The emergence of a reusable innovative BCFC will help to re-address the balance and hence limit the accumulation of contaminated wastage from disposable face coverings.”

Is the tested material safe for the environment and how should it be disposed of?

Line.296.

„However, the filtration efficiencies for fabrics can range from 5 to 80% for particle sizes of >300 nm”

I suggest supplementing the information on the effectiveness with the comparison with currently available mask.

  1. line

”Also, gaps caused by an improper fit of the mask, can result in over a 60% decrease in the filtration efficiency „

Do the proposed masks affect their proper wearing?

  1. line

”The authors have in addition positively evaluated the “Shield” for skin irritation and skin sensitization using a human repeat insult patch test (BioScreen Testing Services, Torrance, California).”

I did not find these results in the "results" section. I highly recommend supplementing this part or making an indication of the source.

Do the authors consider any other side effects of inhaling protein particles (bovine lactoferrin)

to the respiratory system while breathing through the proposed mask?

  1. line

”While the exact mechanism for inhibition of SARS-CoV-2 by lactoferrin is unknown, it has been shown to inhibit viral entry for related-virus SARS- CoV either through virus-binding or cell surface molecule binding”

I suggest defining which cells and in what mechanism more precisely.

”In clinical trials where lactoferrin is used as an antiviral, it has been shown to reduce the severity and duration of disease in patients with SARS-CoV-2 and has been used as a preventative against the contraction of SARS-CoV-2”

I suggesting a more detailed explanation: how applied, how effective, etc.

In the introduction, the authors write about the high cost of masks.

It would be important to know the expected costs in relation to currently available masks.

Round 2

Reviewer 3 Report

I consider the replies and changes to the manuscript sufficient. I have no more comments on the article.